# Qualifying Knowledge and Knowledge Sharing in Multilingual Models

## Abstract

Pre-trained language models (PLMs) have demonstrated a remarkable ability to encode factual knowledge. However, the mechanisms underlying how this knowledge is stored and retrieved remain poorly understood, with important implications for AI interpretability and safety. In this paper, we disentangle the multi-faceted nature of knowledge: successfully completing a knowledge retrieval task (e.g., *"The capital of France is __"*) involves mastering underlying concepts (e.g., *France, Paris*), relationships between these concepts (e.g., *capital of*), the structure of prompts, including the language of the query. We propose to disentangle these distinct aspects of knowledge and apply this typology to offer a critical view of neuron-level knowledge attribution techniques. For concreteness, we focus on Dai et al.'s (2022) Knowledge Neurons (KNs) across multiple PLMs, testing 10 natural languages and unnatural languages (e.g. Autoprompt). Our key contributions are twofold: (i) we show that KNs come in different flavors, some indeed encoding entity level concepts, some having a much less transparent, more polysemantic role , and (ii) we uncover an unprecedented overlap in KNs across up to all of the 10 languages we tested, pointing to the existence of a partially unified, language-agnostic retrieval system. To do so, we introduce and release the Multi-ParaRel dataset, an extension of ParaRel, featuring prompts and paraphrases for cloze-style knowledge retrieval tasks in parallel over 10 languages.

## 1 Introduction

Recent advances in Large Language Models (LLMs) have led to models trained on vast and diverse linguistic datasets drawn from across the Internet, incorporating numerous languages simultaneously (Scao et al., 2023; Touvron et al., 2023; Achiam et al., 2024). However, these languages are not evenly represented, and performance on low-resource languages often depends on cross-linguistic transfer from high-resource languages (Pires et al., 2019; Lample & Conneau, 2019; Conneau et al., 2020a; Huang et al., 2021). Whether LLMs can develop common, language-agnostic representations that enable such zero-shot transfer remains an open question in the literature (Singh et al., 2019; Kudugunta et al., 2019; Kassner et al., 2021). Kervadec et al. (2023) extended this investigation to machine-generated languages, revealing that different representations can emerge, suggesting multiple ways knowledge may be encoded in LLMs.

Understanding how Pre-trained Language Models (PLMs) store and retrieve knowledge is essential for enhancing interpretability and safety in AI systems. Many recent studies have sought to localize and attribute specific knowledge to individual neurons within these models (Dai et al., 2022; Meng et al., 2022; 2023). These methods often attempt to identify neurons whose activations are critical for making accurate predictions. Typically, they focus on neurons in intermediate layers of Feed-Forward Networks (FFNs) within transformer architectures (Geva et al., 2021). These approaches face strong limitations, as highlighted in recent critiques (Hase et al., 2023; Niu et al., 2023; Huang et al., 2023).

In this work, we offer a novel perspective by refining the concept of "knowledge" itself. To correctly complete a prompt like *The capital of France is*, a model must process multiple layers of information: sensitivity to the specific concept *France*, retrieval of the target concept *Paris*, and understanding the relational context *capital of*. We introduce a method to distinguish these subtypes of knowledge—conceptual and relational—that is compatible with any knowledge attribution

Figure 1: The Knowledge Neurons (KNs) hypothesis connects LLM success on a fill-in-the-blank cloze task (e.g. *The capital of France is*) to the activation of a small set of neurons. (a) The same neurons can be selected (green) in response to a single task, thereby qualifying as *concept* neurons (about e.g., Paris) or in response to a range of tasks all concerning a certain relations between concepts, thereby qualifying as *relational neurons* (e.g., *capital of* is a relation between France and Paris, between England and London, etc.). (b) In multilingual LLMs, concept and relational neurons may be selected specifically for a language or across languages.

technique. We apply this method to the Knowledge Neurons (KNs) framework introduced by Dai et al. (2022), to provide a critical view on such a method and extend it to investigate how knowledge is shared across languages in PLMs (Figure 1). Code and data available at [URL redacted for anonymous review].

Our contributions are:

- We propose a finer-grained typology of knowledge, providing a critical perspective on neuron-level attribution methods like the Knowledge Neuron hypothesis, in particular its expectation of monosemanticity.
- We analyze through this prism multiple PLMs (BERT, mBERT, OPT, Llama 2, and Gemma 2), revealing that a substantial number of 'Knowledge Neurons' exhibit polysemantic behavior, while others are specifically responsive to individual concepts or relations.
- We release `Multi-ParaRel`, a multilingual version of the `ParaRel` dataset (Elazar et al., 2021a), which includes 10 languages and is compatible with autoregressive models.
- We demonstrate that LLMs store knowledge in similar neurons across 10 languages, and even in machine-generated languages (AutoPrompt), suggesting a shared cross-linguistic mechanism for knowledge retrieval.

## 2 RELATED WORK

**Multilingual Language Models** Training separate models for different languages is resource-intensive, data-hungry, and generally ineffective at leveraging cross-linguistic similarities and knowledge. In practice, recent LLMs (Touvron et al., 2023; Achiam et al., 2024) are trained on extensive portions of the Internet, making them *de facto* multilingual. Examples include mBERT (Devlin et al., 2019), XLM-R (Lample & Conneau, 2019; Conneau et al., 2020a), mBART (Liu et al., 2020), mT5 (Xue et al., 2021), and BLOOM (Scao et al., 2023), along with their fine-tuned variants like BLOOMZ, mT0 (Muennighoff et al., 2023), and FLAN-T5 (Chung et al., 2022).

The performance of these models is believed to stem from the emergence of efficient representations that are shared across languages (Aharoni et al., 2019; Arivazhagan et al., 2019; Conneau et al., 2020b). Research has investigated their cross-linguistic capabilities using artificial languages (Ri & Tsuruoka, 2022; Deshpande et al., 2022; Guerin et al., 2024), evaluating their performance on tasks across different languages (Pires et al., 2019; Wu & Dredze, 2019), analyzing their translation capabilities (Lample et al., 2018; Sennrich et al., 2016; Artetxe et al., 2018), and assessing their performance on low-resource languages (Garcia et al., 2021), as well as examining their architectural properties (K et al., 2020). Some studies, including the current work, have directly compared representations from one language to another (e.g., using Canonical Correlation Analysis across layers, as in Singh et al., 2019; Kudugunta et al., 2019). The conclusions drawn from these comparisons are mixed. For instance, Singh et al. (2019) argue that representations are distinctly partitioned between languages, while Kudugunta et al. (2019) suggest that representations are more or less shared, depending on the linguistic proximity of the languages.

More directly to the current work, Chen et al. (2024) recently looked at the overlap between knowledge neurons obtained for English and Chinese. We examine a similar overlap, albeit comparing 10 natural languages at once and we believe it is essential. Pairwise sharing leaves ambiguity: are neurons shared with all languages, none, or just pairs? Bias toward a dominant language (e.g., English) or chance sharing between two languages is more likely than sharing across 10. Including 10 languages reveals symmetrical roles in neuron sharing, extending beyond language pairs and providing robust evidence of truly multilingual knowledge representation. We also make a comparison with an 'unnatural language' (Shin et al., 2020). Such prompts provide an extreme test for the idea that knowledge could be accessed independently of form: they are not human-readable, and they had been shown to be processed differently by LLMs (Kervadec et al., 2023).

**Knowledge in LLMs** LLMs acquire knowledge by training on extensive corpora (Petroni et al., 2019; Roberts et al., 2020; Safavi & Koutra, 2021). The work by Petroni et al. (2019) introduced LAMA, a dataset designed to evaluate BERT through a fill-in-the-blank cloze task (e.g., *The capital of France is [MASK].*). Subsequent research has built upon LAMA (Jiang et al., 2021), highlighting the limitations of LLMs as knowledge bases (Elazar et al., 2021b; AlKhamissi et al., 2022), while also attempting to enhance their performance (Wei et al., 2021; Petroni et al., 2020). Consequently, research has emerged focusing on localizing and editing knowledge directly within the model (Radford et al., 2017; Lakretz et al., 2019; Bau et al., 2020b; Sinitsin et al., 2020; Mitchell et al., 2021; 2022; De Cao et al., 2021; Santurkar et al., 2021; De Cao et al., 2022; Bau et al., 2020a; Cohen et al., 2023).

In this context, knowledge attribution methods such as ROME (Meng et al., 2022) and MEMIT (Meng et al., 2023) (both employing causal mediation techniques; Vig et al., 2020), along with Knowledge Neurons (Dai et al., 2022) (utilizing an integrated gradient approach; Sundararajan et al., 2017), have been proposed. These methods are predicated on the assumption that neurons within the intermediate layers of transformers' Feed-Forward Networks (FFNs) encode knowledge. However, we align with other studies (Hase et al., 2023; Niu et al., 2023; Huang et al., 2023) that suggest this assumption may be an oversimplification. While certain neurons play a significant role in specific tasks (Lakretz et al., 2019; Manning et al., 2020; Rogers et al., 2020; He et al., 2024), LLM neurons are not necessarily monosemantic; rather, they can serve multiple functions depending on the context and task (Adly et al., 2024). Furthermore, their effectiveness in altering knowledge is subjective and widely debated (Hase et al., 2023). Other works (Wang et al., 2024; Tang et al., 2024; Kojima et al., 2024) have identified multilingual neurons in LLMs; this paper focuses specifically on knowledge-related neurons, offering a more precise analysis. We propose a knowledge-attribution method-agnostic typology, illustrated with Dai et al.'s (2022) Knowledge Neurons. This approach aims to provide a critical view on the Knowledge Neurons hypothesis while exploring what insights it can offer regarding how knowledge is encoded in LLMs.

## 3 METHODOLOGICAL BACKGROUND

**Knowledge** The TREx dataset (Elsahar et al., 2018) is a collection of relational facts stored in triplets of the form $< h, r, t >$, with $r$ a relation and $h$ and $t$ entities entering in that relation. TREx exhibit 41 relations, such as *being the capital of*, *was born in*, etc. Each full triplet can be referred to as an **instantiation** of its own relation $r$.

**Knowledge Localization Methods** Geva et al. (2021) observed that a FFN can be seen as a Key-Value memory system, similar to self-attention. To assess if and where knowledge could be stored in FFNs, Dai et al. (2022) used a knowledge attribution method based on integrated gradients (see next paragraph for details). They show that a fact (e.g., *The capital of France is Paris*) can be associated to a few neurons (around 4), whose activations correlate with the probability of the model to fill in the elements of the fact appropriately. Similarly, Meng et al. (2022) proposed Rank-One Model Editing (ROME), which uses causal mediation to localize and edit knowledge in GPT, and Meng et al. (2023) introduced Mass-Editing Memory in a Transformer (MEMIT), which edits facts at scale. All of these knowledge attribution methods have their limitations; we apply our analysis to the Knowledge Neurons by way of illustration. Our approach is applicable to all such methods.

**Knowledge Neurons**   Dai et al. (2022) track Knowledge Neurons (KNs) during a fill-in-the-blank cloze task (see also Petroni et al., 2019) based on TREx. Let $w_i^{(l)}$ be the $i^{th}$ neuron of the intermediate layer of the $l^{th}$ FFN. The knowledge score of a neuron $w_i^{(l)}$ is calculated through the integrated gradient attribution method (Sundararajan et al., 2017), KNs are then filtered through thresholds. First, they retain only neurons with an attribution score greater than $t_{kn} \times \max_{i,l} \text{Attr}_{h,p_r,t}(w_i^{(l)})$. This procedure is carried out for each prompt associated with a fact $< h, r, t >$, and thus yields a set of candidate KNs per prompt. Let us denote $N_r$ the number of prompts for a given relation $r$. To get results robust to noise, and to factor out signal associated to specific prompts rather than knowledge, they keep only neurons appearing in the candidate neurons set of at least $p_{kn} \times N_r$ prompts. They propose thresholds of $t_{kn} = 0.2$ (only keep neurons scoring at least at 20% of the max attribution score) and $p_{kn} = 0.7$ (only keep neurons appearing in at least 70% of the different prompts for a given relation).

## 4 METHOD

**Datasets**   For relational facts, we used the TREx dataset (Elsahar et al., 2018), which comprises 41 relations with approximately 1,000 facts per relation. For prompts, we employed the augmented version of `ParaRel` provided by Kervadec et al. (2023). This version retains only prompts compatible with autoregressive models and enriches the dataset with multiple paraphrases for each relation. In Section 6, we explore multilingual models, which we tested on the multilingual variant of LAMA (Kassner et al., 2021) as well as on a new multilingual version of `ParaRel` that we introduce. We refer to this new dataset as `Multi-ParaRel`.

The detailed methodology for creating `Multi-ParaRel`, along with a quality assessment, is provided in Appendix A. Our dataset currently spans 10 languages: English, French, Spanish, Catalan, Danish, German, Italian, Dutch, Portuguese, and Swedish. We also investigate an unnatural language: AutoPrompt. Following the same train, development, and test splits as Shin et al. (2020), we trained 10 different seeds of AutoPrompt for each relation and each model. We also make these sets of prompts available.

**Concept Neurons and Relation Neurons**   We propose a simple typology that refines the type of knowledge attributed while answering fill-in-the-blank cloze tasks. For example, correctly answering the question *What is the capital of France?* not only requires knowledge of the answer *Paris*, but also an understanding of the relationship between *France* and *Paris*. We thus introduce a simple principle: a neuron that is hypothesized to encode a specific concept, such as one about *Paris*, should not be also responsible for encoding other concepts, and should therefore not be associated to other facts such as *The capital of Spain is Madrid*. If a neuron consistently encodes the same relation across multiple instances, we refer to it as a relational neuron, indicating that it is sensitive to a relation, such as *capital of*.

We thus define **Relation Neurons** as KNs that appear in at least $t_r \times N$ instances of facts associated with a particular relation, where $N$ is the total number of facts, and $t_r$ is a predefined relational threshold. In contrast, neurons that appear in less than $t_c \times N$ of the facts, for some other threshold $t_c$, are referred to as **Concept Neurons**, as they are more likely to encode specific pieces of knowledge or information about individual entities.

The aim is to test the robustness of this distinction by investigating the role of the thresholds $t_r$ and $t_c$. A 'clean' scenario that supports the Knowledge Neuron hypothesis and the idea of monosemanticity would show that some concept neurons are found even for $t_C \times N = 1$ (very specific to a concept), and relational neurons are found when $t_R \times N = N$ (completely systematically present for a relation). Alternatively, softer boundaries would suggest that these KNs play a more polysemantic and nuanced role, whereby knowledge is partially distributed across different neurons on different occasions (e.g., the concept of *Paris* and *Madrid* cannot be disentangled at the neuron level, or the relation *capital of* is not always encoded in the same way).

As we do not know a priori which neurons play specific roles, we performed an exhaustive study across varying thresholds. In fact, it is part of the method to look at all possible thresholds to identify the behavior of KNs. Moreover, no major variation based on the choice of threshold was found.

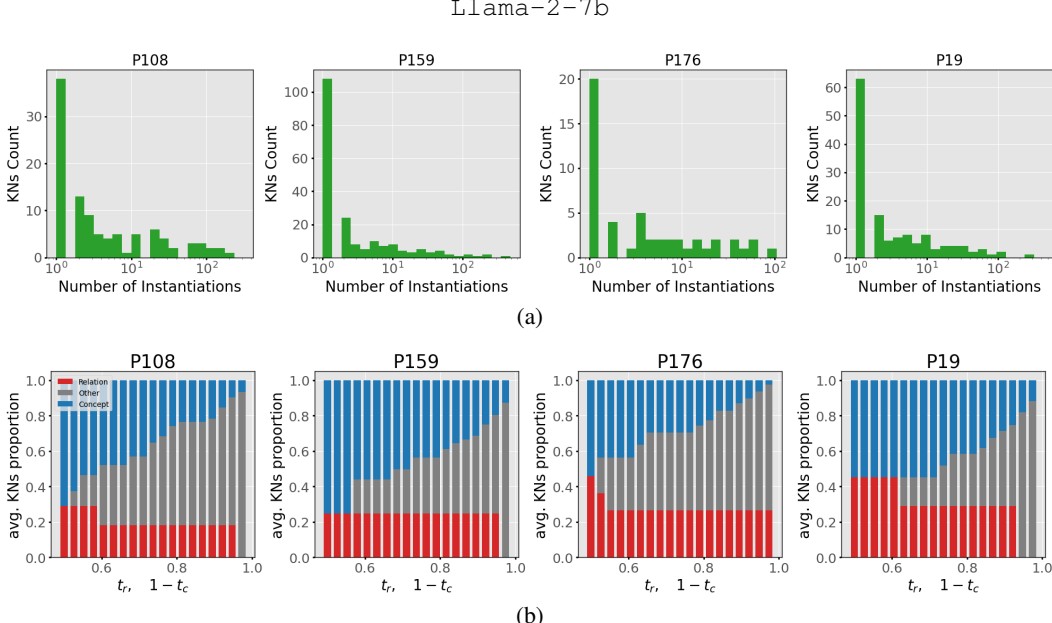

Figure 2: Each panel corresponds to a relation (P108, P159, etc.). (a) Distribution of KNs based on the number of instantiations (i.e. specific triplets, specific facts) within a relation for which a KN was identified. A large number of neurons are identified as KN for a single instantiation, while a roughly similar number of neurons are identified as KN for a continuously increasing number of instantiations within a relation. (b) Average proportion of the KNs from a single instantiation which can be categorized as **Concept**, **Relation** or neither, according to different thresholds (x-axis). The proportion of relational neurons is stable across different thresholds, the proportion of concept neurons decreases with more demanding thresholds.

**Multilingual Knowledge Neurons** Similarly, we ask whether knowledge is language-agnostic; for example, humans do not need to relearn facts when acquiring a new language. Knowledge could be language-dependent in LLMs however: if a fact is present from the English corpus but missing from a Spanish training corpus, an LLM may be able to retrieve that knowledge when prompted in English but not when prompted in Spanish. We employ the KNs framework to investigate the open question of whether a common language-agnostic knowledge representation exists in multilingual models at the level of neurons.

We hypothesize that some KNs may be specific to one language, while others may be sensitive to prompts in multiple languages. We thus analyze the number of languages across which such neurons are shared. We do so by identifying KNs for relations in the `ParaRel` dataset across multiple languages, using the `Multi-ParaRel` dataset, which was specifically created for this multilingual evaluation.

# 5 MONOLINGUAL EXPERIMENTS: TRACKING CONCEPT AND RELATION NEURONS

## 5.1 EXPERIMENTAL SETTINGS

**Models** We studied BERT (Devlin et al., 2019), and more precisely `bert-base-uncased` and `bert-large-uncased`, as it has been the reference model for evaluation on TREx since Petroni et al. (2019). Having been trained on Wikipedia from which TREx is derived, their performance is very good (P@1> 0.4). We also studied OPT (Zhang et al., 2022) in its 350 million-parameters version `opt-350m` and 6.7 billion-parameters version `opt-6.7b`, Llama 2 (Touvron et al., 2023) in its 7 billion-parameter version `Llama-2-7b-hf` as well as Gemma 2 (Team et al., 2024) in its

9 billion parameters version gemma-2-9b. For all these models we use the HuggingFace implementation. KNs computations were performed on NVIDIA Tesla V100 GPUs for models with less than a billion parameters, and on NVIDIA Tesla A100 GPUs for larger models. The computation took less than an hour per relation.

**Template filtering**   Per model, we excluded prompts with less than 10% top-1 accuracy (that is, accuracy of the most probable continuation). We then excluded relationships with less than 4 prompts left. Since all actual answers were made of a single token, we also limit answers made of a single token. After this filtering, we obtained on average 15 prompts per relation for BERT and 8 prompts per relation for OPT (starting from 18), confirming the higher accuracy of BERT at the task.

## 5.2 TRACKING A TYPOLOGY OF KNOWLEDGE

Before classifying Knowledge Neurons (KNs) according to our typology, we first analyzed the distribution of KNs based on the number of instantiations for which a KN was identified. Figure 2a illustrates the results for four relations using the Llama-2-7b model (complete results are provided in Appendix B). A qualitative analysis reveals two key findings: (i) many KNs appear in only one instantiation, indicating that these neurons are task-specific and sensitive to a single concept; and (ii) there is a continuous range of KNs sensitive to between 3 and $N$ instantiations, suggesting a more nuanced role for these neurons that lies between relational and conceptual.

The second observation challenges the simplistic interpretation of assigning neurons exclusively to concepts. At the same time, it also demonstrates the presence of a significant number of neurons sensitive to enough instantiations to hypothesize a more relational role in knowledge retrieval mechanisms.

Thus, we have identified potential candidates for the roles of both **Concept Neurons** and **Relation Neurons**, as well as neurons that fall into an intermediate category. The natural question that arises is: what is the proportion of each neuron type per instantiation, based on thresholds $t_r$ and $t_c$? This information is not directly inferable from Figure 2a, as neurons appearing consistently across instantiations are less visible than neurons that appear uniquely in each instance.[1] To address this, we computed the proportion of each neuron type as a function of thresholds at the instantiation level (see Figure 2b). For simplicity, we used symmetrical thresholds, setting $t_r = 1 - t_c$.

As expected, when the thresholds become more restrictive, the number of neurons with well-defined roles decreases, giving way to neurons with less clearly defined functions across all relations. For the Llama-2-7b model, we observe that the number of neurons classified as **Relation Neurons** remains more stable compared to those classified as **Concept Neurons**. Furthermore, for a single instantiation, there are few KNs that are exclusive to that instance: when $t_c < 0.1$, the proportion of **Concept Neurons** is less than 0.2.

We also examined the distribution of neuron types across the model's layers but found no significant variation. As observed by Dai et al. (2022), KNs are primarily concentrated in the final layers.

In summary, we have demonstrated the existence of neurons reacting specifically to a single concept within a relation. We have also identified neurons that play a much broader role in such relations, with some reacting to almost all instances of that relation. We attempt to verify this hypothesis through causal experiments in the next section. Finally, some neurons are activated by a subset of the instantiations, carrying a much less transparent type of knowledge. In principle, it could encode subtypes of relations, such as 'capital of a European country', although we find this highly stipulative at the moment. In the next section, we will focus on concept and relation neurons and evaluate their role through causal experiments.

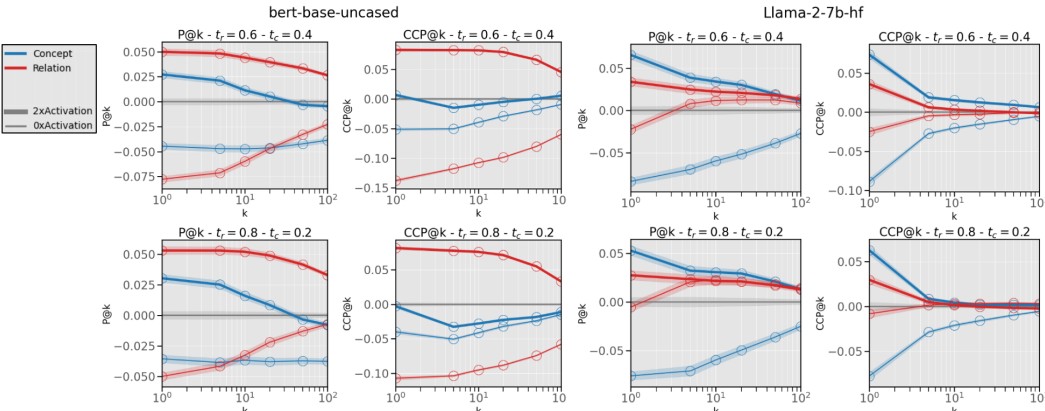

Figure 3: Boosting experiments results for `bert-base-uncased` (left) and `Llama-2-7b` (right) for two couple of thresholds $t_r = 0.6$, $t_c = 0.4$ (top) and $t_r = 0.8$, $t_c = 0.2$ (bottom). The lines corresponds to the $\Delta$P@k (resp. $\Delta$CCP@k) for different $k$ values ranging from 1 to 100. Thick lines represents the doubled activations results and thin lines the nullified activations results. We also plotted the standrad error accross the evaluated instantiations of the relations.

## 5.3 BOOSTING EXPERIMENTS

In this experiment, we investigate the effect of either doubling or nullifying the activation of KNs on model predictions. Dai et al. (2022) conducted similar experiments, focusing on how manual changes to neuron activations influenced output probabilities. In contrast, we employ two more concrete impact metrics: precision at rank k, denoted P@k, which measures the proportion of correct responses in the top k model predictions, and correct category proportion at rank k, denoted CCP@k, which reflects the proportion of responses in the correct category (e.g., *capitals*) within the top k predictions. The original metric of relative probabilities change would not show specificity (e.g. unrelated tokens could be even more boosted). For this reason, we report P@k and CCP@k. Effects here ensure that the boost to the correct answer overcomes any boost for other answers. We also include a control experiment in Appendix B to better investigate specificity.

Our goal is to verify whether the behavior of the identified KNs aligns with our proposed typology. Specifically, we hypothesize that (i) there will be a marked increase (or decrease) in precision at rank k=1 when the activations of **Concept Neurons** are doubled (or nullified), with the effect diminishing as k increases. Similarly, we anticipate (ii) that the effect of **Relation Neurons** on P@k will be weaker than that of **Concept Neurons**, as precision is primarily sensitive to the correct response. In contrast, for the CCP@k metric, we expect (iii) that **Relation Neurons** will play a more significant role, as these neurons should be more likely to favor the correct category (e.g., *capitals*), even if it does not boost the correct answer specifically. We assess these effects for a range of thresholds $t_c$ and $t_r$. Results for the `bert-base-uncased` and `Llama-2-7b` models are shown in Figure 3 (see Appendix B for additional models and thresholds as well).

The figures show the delta in P@k and CCP@k for the predictions with altered (doubling or nulifying) vs unaltered activations. The horizontal line at zero thus represents the baseline model performance. Of the six models evaluated, all six display the expected effect (i) consistently across all thresholds: in short, the top response is more accurate when the activations of concept neurons are increased. However, only two models, Llama-2 and the Gemma-2, exhibit effect (ii). Additionally, four models, belonging to the BERT and OPT families, align with expectation (iii). Overall, `bert-large-uncased` and `gemma-2-9b` adhere to all three expected behaviors across all cases. This happens under restrictive thresholds however ($t_r = 0.9$ and $t_c = 0.1$), and the four other tested models fail to match all of these expectations.

---

[1]For example, if each instantiation contains 10 KNs, including 2 perfect conceptual neurons and 8 perfect relational neurons (present in only 1 instantiation and all instantiations, respectively), Figure 2a would display a bar of 200 at the 1 abscissa and a bar of 8 at the 100 abscissa, which would obscure the predominant role of **Relation Neurons**.

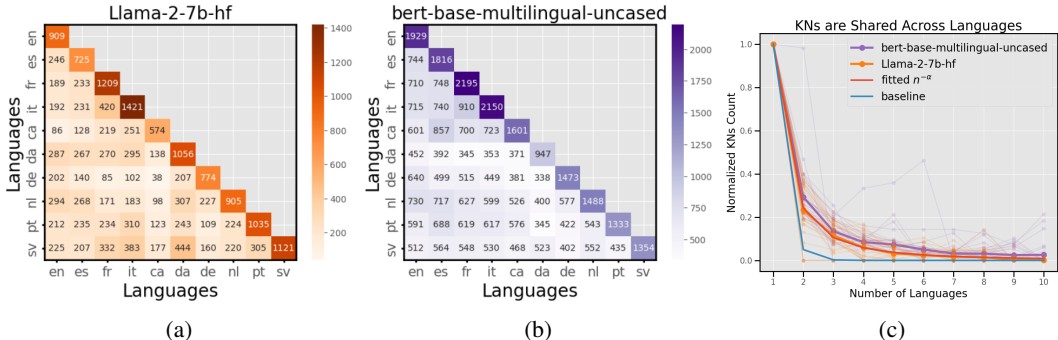

Figure 4: (a) Number of KNs shared by language pairs for `Llama-2-7b`. About a quarter of neurons are shared between two languages. (b) Same for `bert-base-multilingual-uncased`. (c) Proportion of shared KNs in a relation as a function of the number of languages in the intersection for `Llama-2-7b` and `bert-base-multilingual-uncased`.

These mixed results show that classifying KNs into distinct and disentangled roles is not perfect, potentially due to noise in our methods or in knowledge attribution methods in the first place. Yet, our experiments do indicate that, for certain models, KNs exhibit specific behaviors and manipulating them leads to predictable effects.

## 5.4 DISCUSSION

As anticipated, these experiments underscore the complexity of the internal mechanisms within LLMs, making it impractical to map a single, well-defined function to individual neurons. Many of the identified KNs do not adhere to a clearly defined role and cannot be neatly categorized as encoding either concepts or relations, even within a highly controlled environment like `ParaRel`. We believe that the polysemantic nature of neurons prevents such precise delineation, which also helps explain the knowledge editing limitations highlighted in prior research. However, contrary to our initial expectations, certain KNs do appear to serve rather specific functions, and this has been experimentally confirmed for some models in boosting experiments. Hence, while the idea that knowledge would be represented entirely in mono-semantic single neurons is unrealistic, the historically associated methods of, e.g., Knowledge Neurons nonetheless detect transparent signal about how knowledge is encoded. KNs are thus a useful tool to pursue the study of knowledge representation in multilingual models too, which we do in the next section.

## 6 MULTILINGUAL EXPERIMENTS

When we learn a new language, we do not learn all facts about the world again, just new ways to express them. That is, there is a central knowledge base, that we can prompt with several languages. In this section we inquire if knowledge is shared across languages in multilingual models too and, if so, what knowledge.

### 6.1 EXPERIMENTAL SETTINGS

**Models** For this experiment we studied `bert-base-multilingual-uncased` (Devlin et al., 2019) and `Llama-2-7b`. We used a NVIDIA Tesla V100 GPU for BERT and NVIDIA Tesla A100 GPU for Llama 2, both for about one hour per relation and per language.

**Multi-ParaRel** We built and release a new dataset `Multi-ParaRel`, a multilingual version of `ParaRel`. More details are given in Appendix A. `Multi-ParaRel` currently includes 10 languages: English, French, Spanish, Catalan, Danish, German, Italian, Dutch, Portuguese and Swedish. We also offer a translation and curation pipeline which makes it possible to add more paraphrases and more languages. It has an average of 17 prompts per relation and per language but this value varies (from 9 for German to 19 for English). Each prompt is compatible with autoregres-

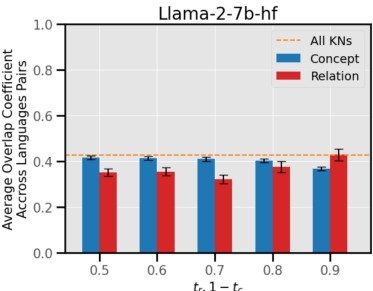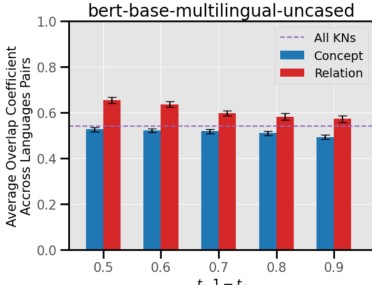

Figure 5: Influence of typology on the average overlap coefficient calculated per language pair of `Llama-2-7b` (left) and `bert-base-multilingual-uncased` (right).

sive models. After filtering for quality as above, we obtain on average 10 prompts per relation and language.

## 6.2 Knowledge Neurons are Shared Across Languages

**Are KNs Bilingual?**  KNs were calculated separately for each relation and language. A KN is considered shared between two languages if it appears as a KN in both languages for the same relation. We conducted this pairwise analysis across all languages, thereby extending the findings of Chen et al. (2024) to encompass 10 languages.

The results are presented in Figures 4a and 4b. For the `Llama-2-7b` model, over a quarter of the neurons are shared between any two languages, with this proportion increasing to approximately one-third for `bert-base-multilingual-uncased`. This represents a significant degree of neuron sharing, especially when considering that `bert-base-uncased`, for example, has more than $12 \times 3,072 = 36,864$ neurons in the intermediate layers of its FFNs. To quantify this, note that among these $36,864$ neurons, only $1,929$ are identified as KNs across all relations for English, and $2,195$ for French (roughly $5\%$). If KNs were randomly selected for each language, we would expect around 100 shared neurons between them ($5\%$ overlap); however, in reality, 710 neurons are shared. A similar analysis for `Llama-2-7b` gives even more extreme results: by chance, there should be 2 shared neurons, while in practice 189 are found. Moreover, these numbers represent a lower bound, as some relations were excluded from the prompt filtering process for certain language pairs, effectively reducing the shared KN count for those relations to zero. Thus, the data indicates significant overlap of KNs across languages, suggesting a partially shared mechanism for knowledge retrieval across different language pairs.

**Are KNs Multilingual?**  Next, we examine how the number of shared KNs scales with the number of languages in the intersection. Figure 4c shows these results for all relations, along with the average behavior. Across all relations, we observe a consistent pattern: the number of shared neurons decays as a function of the form (number of languages)$^{-\alpha}$, with a fitted $\alpha = 2.04$ for `Llama-2-7b`. In comparison, if neurons were shared at random, the expected behavior would follow $\propto p^{\text{number of languages}}$, where $p$ is the probability of a neuron being a KN (e.g. $p = 0.05$ for BERT). This demonstrates that KNs are more multilingual than chance, reinforcing the notion of a language-agnostic knowledge retrieval mechanism. Similar to the findings in Section 5, we observe some but few neurons activated for *all* languages.

**Are some neurons more Multilingual?**  **Concept Neurons** and **Relation Neurons** were computed separately for each language and each model. Figure 5 displays the average pairwise overlap coefficient for each neuron type, across various $t_r$ and $t_c$ thresholds, alongside with the pairwise overlap coefficient for all KNs. The results reveal a significant difference in overlap between **Concept Neurons** and **Relation Neurons** at all threshold levels. However, the direction of this difference varies depending on the model and on the threshold. At the most demanding thresholds (those to the right selecting the purest types), we observe that relational neurons appear to be more bilingual. Given the variability at other thresholds (in particular for Llama, which is less performant than BERT in this task), we remain cautious about this conclusion.

### 6.3 KNOWLEDGE NEURONS ARE SHARED BETWEEN NATURAL AND UNNATURAL LANGUAGES

We have extended the analysis to non-natural languages, in order to deepen the work of Kervadec et al. (2023) in the specific framework of KNs. More specifically, we calculated 10 seeds of Autoprompt (Shin et al., 2020) for each model and each relation of `ParaRel` and the associated KNs. We then calculated the overlap coefficient between the KNs calculated in this way and those calculated for English at the relationship level. The results are presented in Table 1. This reveals a very large overlap for all models, going up to an almost complete overlap ($\geq 80\%$) for models other than BERT. In the same way that there were important overlaps across natural languages, this new result suggests a similar mechanism of knowledge retrieval even between natural and non-natural languages. It is possible however that there exists a confound here because both Autoprompt and KNs are gradient based.

| Model | bert-base | bert-large | opt-350m | opt-6.7b | Llama-2-7b |
|---|---|---|---|---|---|
| **Avg. Overlap Coeff.** | 40% | 32% | 83% | 87% | 79% |

Table 1: Average overlap coefficient of KNs sets computed at the relation level between English and Autoprompt.

## 7 DISCUSSION

While knowledge neurons may be shared across languages, this does not guarantee that they serve the same role in the two languages. A neuron active in both English and French for a given task may perform different overall tasks depending on the language, that is, parallel activation does not equate to shared functionality. Here, we partially controlled for this and narrowed down the role of these neurons by computing intersections across languages and at the relation level. Yet, further work is needed to investigate more intersections, narrowing down the possible roles, at the level of relations, concepts, responses or formats of the prompt. Our method can further help narrow down shared functions, across languages.

## 8 CONCLUSION

We introduced a typology for knowledge and applied it to the knowledge attribution method proposed by Dai et al. (2022) to better classify and understand the behavior of Knowledge Neurons (KNs). Notably, our method remains agnostic to the specific knowledge attribution technique used. Coherently with the initial assumptions in the original work, we found that some of these neurons encode specific concepts, but we also found many which do not and instead seem to exhibit a distributed role, where multiple neurons share responsibility for encoding concepts within the same relation, or maybe encode the whole relation. We hypothesize that this polysemantic nature of neurons contributes to the mixed success observed when using KNs for knowledge editing tasks. Yet again, we were able to identify a subset of more specialized neurons, which we categorized as either conceptual (sensitive to a single concept) or relational (sensitive to relationships between concepts). And in some contexts their manual manipulations show the expected effects on downstream tasks. We extended our analysis to multilingual models and found that a significant number of KNs are shared across languages—both in pairwise comparisons and across all 10 languages tested. This indicates the presence of a shared, language-agnostic knowledge base within multilingual models. To facilitate this research, we created a multilingual dataset of facts and prompts, enriched with paraphrases in 10 languages. Our findings suggest that even a simple method like Knowledge Neurons can provide valuable insights into the benefits of multilingual training. Looking ahead, we aim to further explore how this shared knowledge can be leveraged to improve the integration of new languages into existing multilingual models. Our results indicate that it may not be necessary to relearn factual knowledge for each language, which could pave the way for more efficient training strategies, particularly for low-resource languages. Instead of focusing on exhaustive coverage of world knowledge, future efforts could prioritize data that highlights the unique syntactic and linguistic features of these languages, thus optimizing resource use and improving model performance.

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

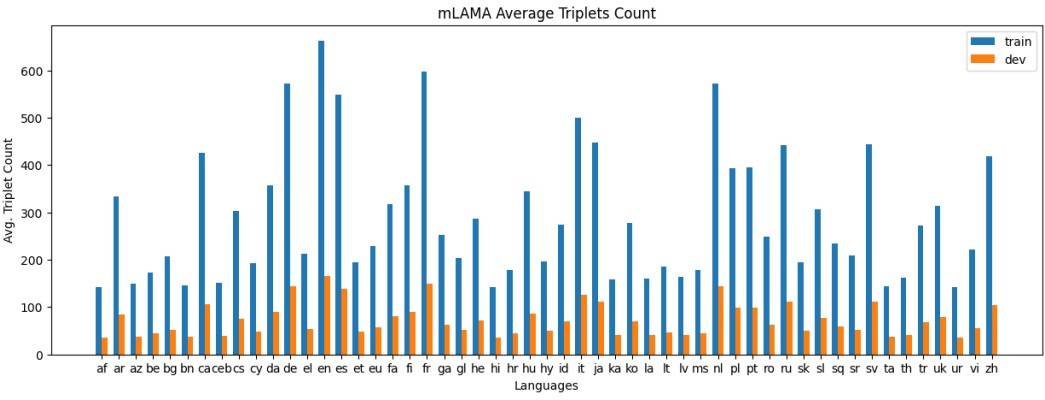

Figure 6: In mLAMA, the number of triplets available varies widely across the different languages.

## A NEW DATASET: MULTI-PARAREL

**Creation Procedure** To build the `Multi-ParaRel` dataset, we used our augmented autoregressive version of `ParaRel` and mLAMA. The goal is to translate a template such as *The capital of [X] is [Y]*. The problem is that translators are confused by the presence of placeholders *[X]* and *[Y]*, often resulting in translation errors. To overcome the difficulty, we instantiated *[X]* and *[Y]*, translated the whole sentence with these specific instances, and replaced the instantiations back with placeholders. To do so, we used mLAMA, which contains triplets for over 53 languages.

For example, consider the translation from English into French of the template:

*The capital of [X] is [Y]*

We use the English triplet <*Great Britain, capital of, London*> to obtain the sentence:

*The capital of Great Britain is London*

This sentence is then translated into French:

*La capitale de la Grande Bretagne est Londres*

Then using the French version of the original triplet (<*la Grande Bretagne, capital of, Londres*>), we can find and replace the entity elements of the triplet with placeholders *[X]* and *[Y]*, resulting in the new template:

*La capitale de [X] est [Y]*

With this overall idea, we can now provide more detail. First of all, such a protocol requires associated triplets in mLAMA from one language to another. However, mLAMA has many more triplets in English than in other languages (see Figure 6), and some triplets are language-specific and therefore cannot be associated with triplets in other languages. We therefore looked into a common English-Target language subset. Then, to avoid translation errors, problems linked to gendered determinants and redundancy (two different templates in English but translated identically in the target language), we used a voting system. Each template was translated 30 times, using 30 triplets. Each translation is assigned a score, which is the number of times the template has been obtained out of the 30 triplets. The template with the highest score is then retained, provided that (i) it is autoregressive, (ii) it has not already been selected and (iii) it is in the top 5 translations.

As a translation model, we used Meta's SeamlessM4T and, more specifically, the Huggingface implementation[2]. We used an NVIDIA Tesla V100 GPU for inference.

**Statistics and Exemples** Table 3 provides examples of translated templates from different languages and relations. The average number of templates obtained per relationship for each language is:

---

[2]https://huggingface.co/facebook/seamless-m4t-large

| Language | Avg templates |
|---|---|
| Catalan | 19 |
| Danish | 15 |
| Dutch | 17 |
| English | 19 |
| French | 14 |
| German | 9 |
| Italian | 19 |
| Portuguese | 19 |
| Spanish | 19 |
| Swedish | 16 |

Table 2: Language Values

| Relation | English | Spanish | French |
|---|---|---|---|
| P36 | *The capital of [X] is [Y]* | *La capital de [X] es [Y]* | *La capitale de [X] est [Y]* |
| | *[X], which has the capital [Y]* | *[X], que tiene la capital [Y]* | *[X], dont la capitale est [Y]* |
| P106 | *The occupation of [X] is [Y]* | *La ocupación de [X] es [Y]* | *La profession de [X] est [Y]* |
| | *[X] works as [Y]* | *[X] trabaja como [Y]* | *[X] travaille comme [Y]* |
| P1001 | *[X] counts as a legal term in [Y]* | *[X] cuenta como término legal en [Y].* | *[X] est un terme légal en [Y]* |
| | *[X] is a valid legal term in [Y]* | *[X] es un término legal válido en [Y].* | *[X] est un terme juridique valide en [Y]* |

Table 3: Examples of templates from `Multi-ParaRel`

**Quality Analysis** To judge the quality of our dataset, we asked a native speaker of French and a native speaker of Spanish to rate the resulting templates in three categories: fluent, weird, ungrammatical. For French 88% are correct, 7% weird and 5% are ungrammatical. For Spanish: 78% of sentences are fluent, 10% weird and 12% are ungrammatical. Although imperfect, `Multi-ParaRel` coupled with a less efficient filtering of prompts gives very good results on mLAMA.

## B  FULL RESULTS

First we provide an overview of all the models behavior with respect to our expectations in Table 4. We also add a control experiment for the BERT family where we conducted the same boosting experiments but sampling KNs randomly within the relation for the Concept Neurons and across relations for Relation Neurons. The goal of such a control is to test the specificity of identified KNs. Results are in Table 5. We see that the effects are destroyed when looking at randomly selected KNs.

Second, we provide all graphs computed for all models and relations concerning the distinction between concept and relation neurons. This corresponds to the results as presented in Section 5.2, Figure 2, also showing all relations each time. Second, we provide all graphs corresponding to the boosting experiments (Section 5.3, Figure 3).

| Model | Expectation (i) | Expectation (ii) | Expectation (iii) |
|---|---|---|---|
| bert-base-uncased | Yes | No | Yes |
| bert-large-uncased | Yes | Yes | Yes |
| opt-350m | Yes | No | Yes |
| opt-6.7b | Yes | No | Yes |
| Llama-2-7b | Yes | Yes | No |
| gemma-2-9b | Yes | Yes | No |

Table 4: Overview of boosting results for all models. Expectations are: (i) there will be a marked increase (or decrease) in precision at rank k=1 when the activations of **Concept Neurons** are doubled (or nullified), with the effect diminishing as k increases, (ii) the effect of **Relation Neurons** on P@k will be weaker than that of **Concept Neurons**, as precision is primarily sensitive to the correct response, (iii) **Relation Neurons** will play a more significant role, as these neurons should be more likely to favor the correct category (e.g., *capitals*), even if it does not boost the correct answer specifically.

| Model | Expectation (i) | Expectation (ii) | Expectation (iii) |
|---|---|---|---|
| bert-base-uncased | No | No | Yes but effect $10\times$ smaller |
| bert-large-uncased | No | No | No |

Table 5: Overview of boosting results for the control experiment.

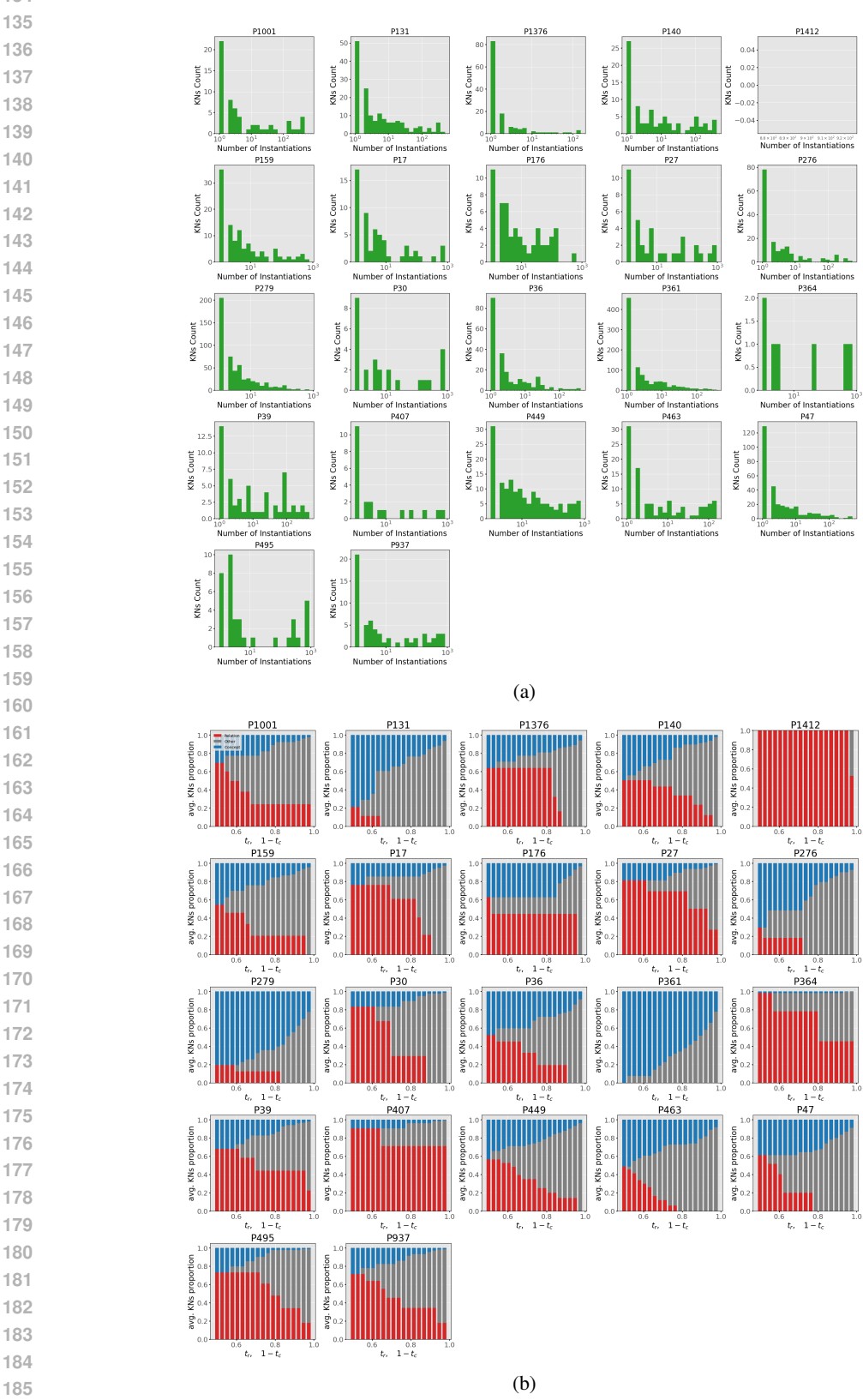

Figure 7: `bert-base-uncased`

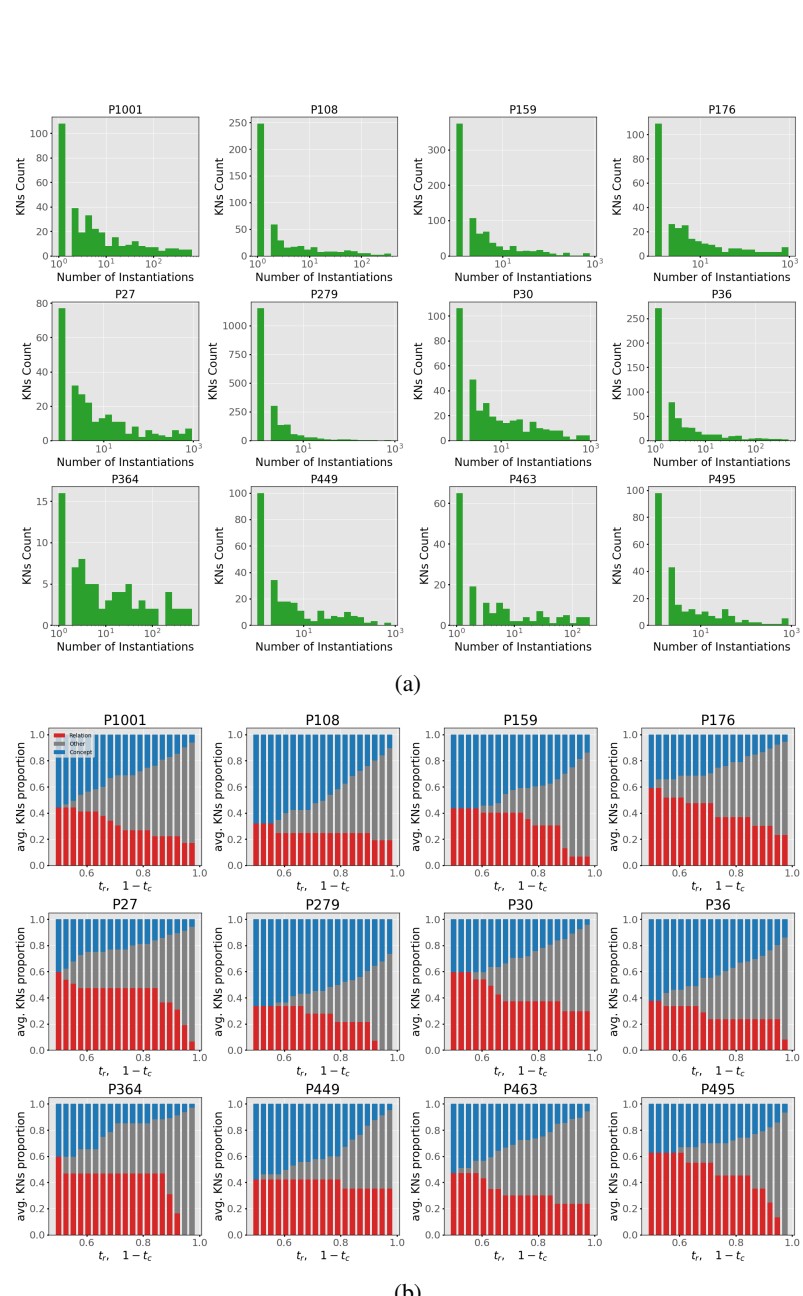

Figure 8: `opt-6.7b`

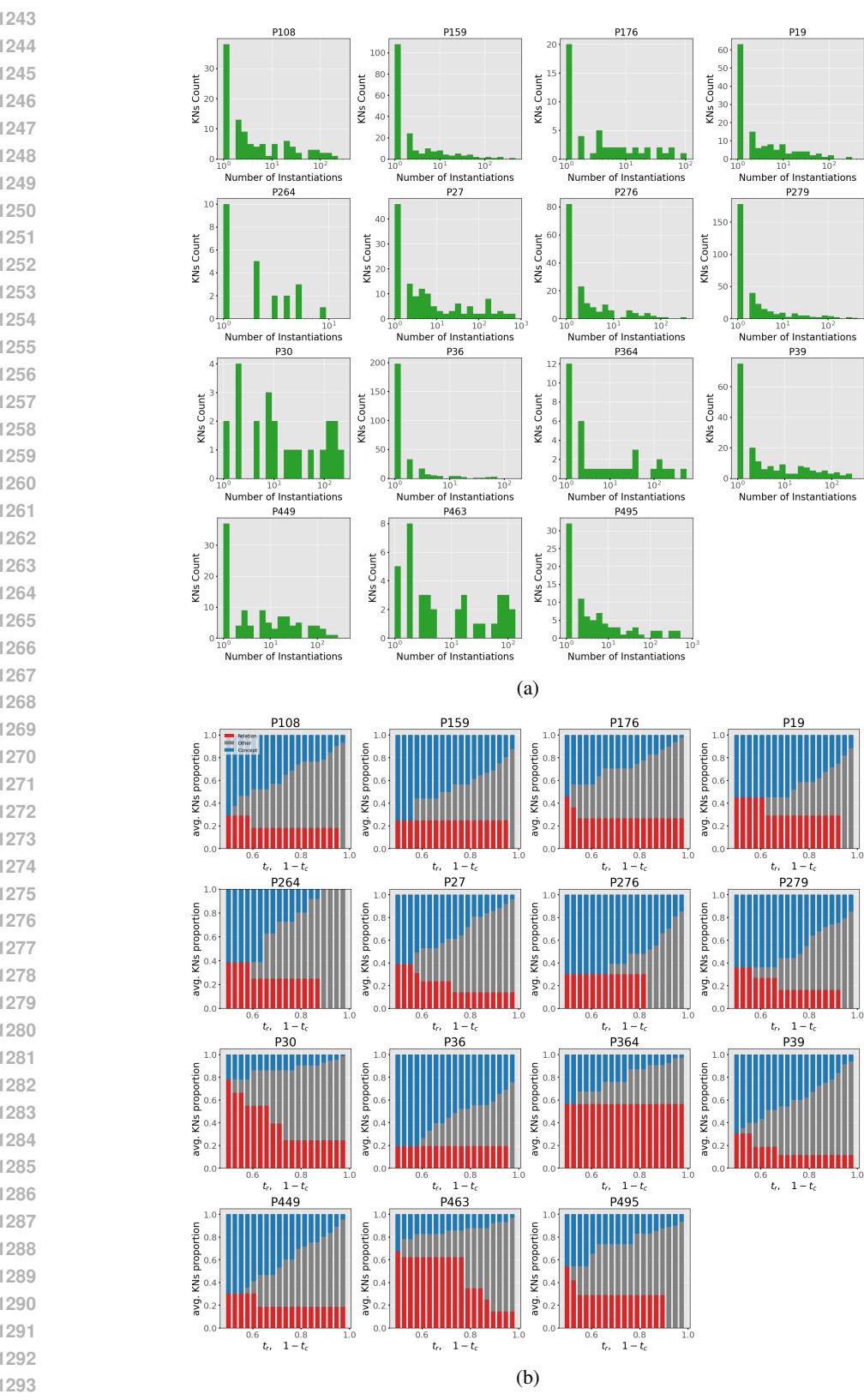

Figure 9: `Llama-2-7b`

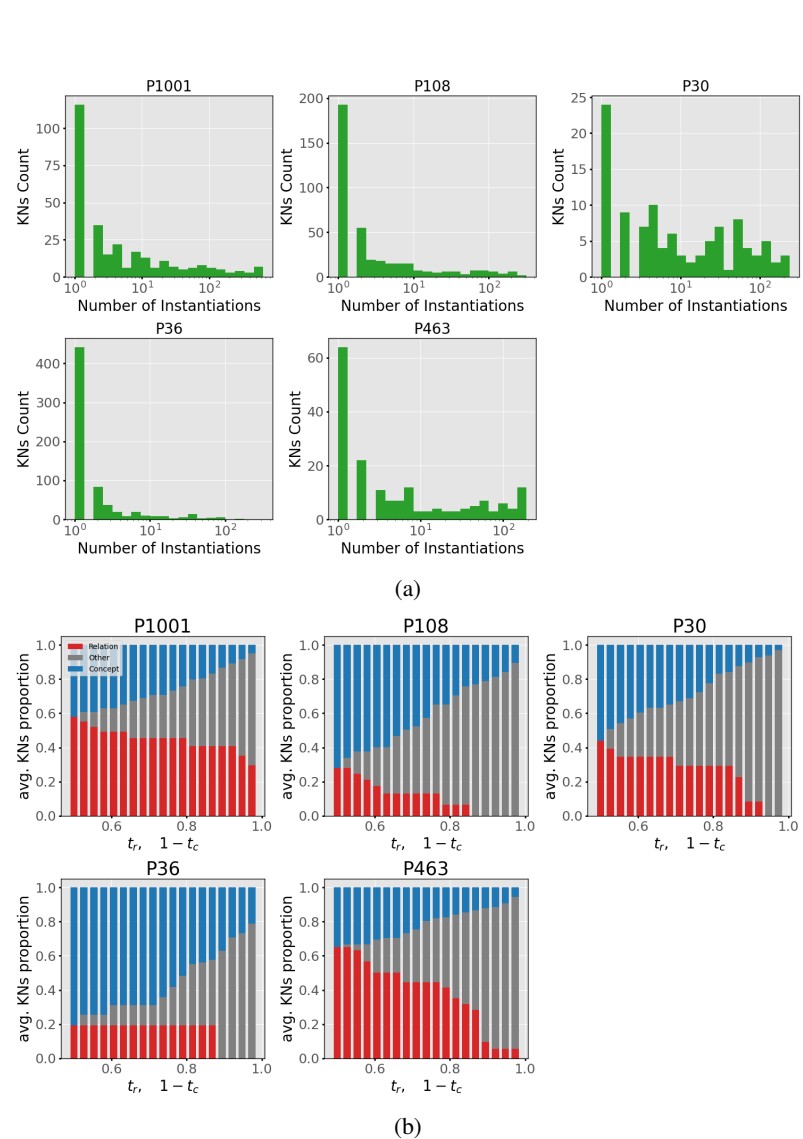

Figure 10: gemma-2-9b

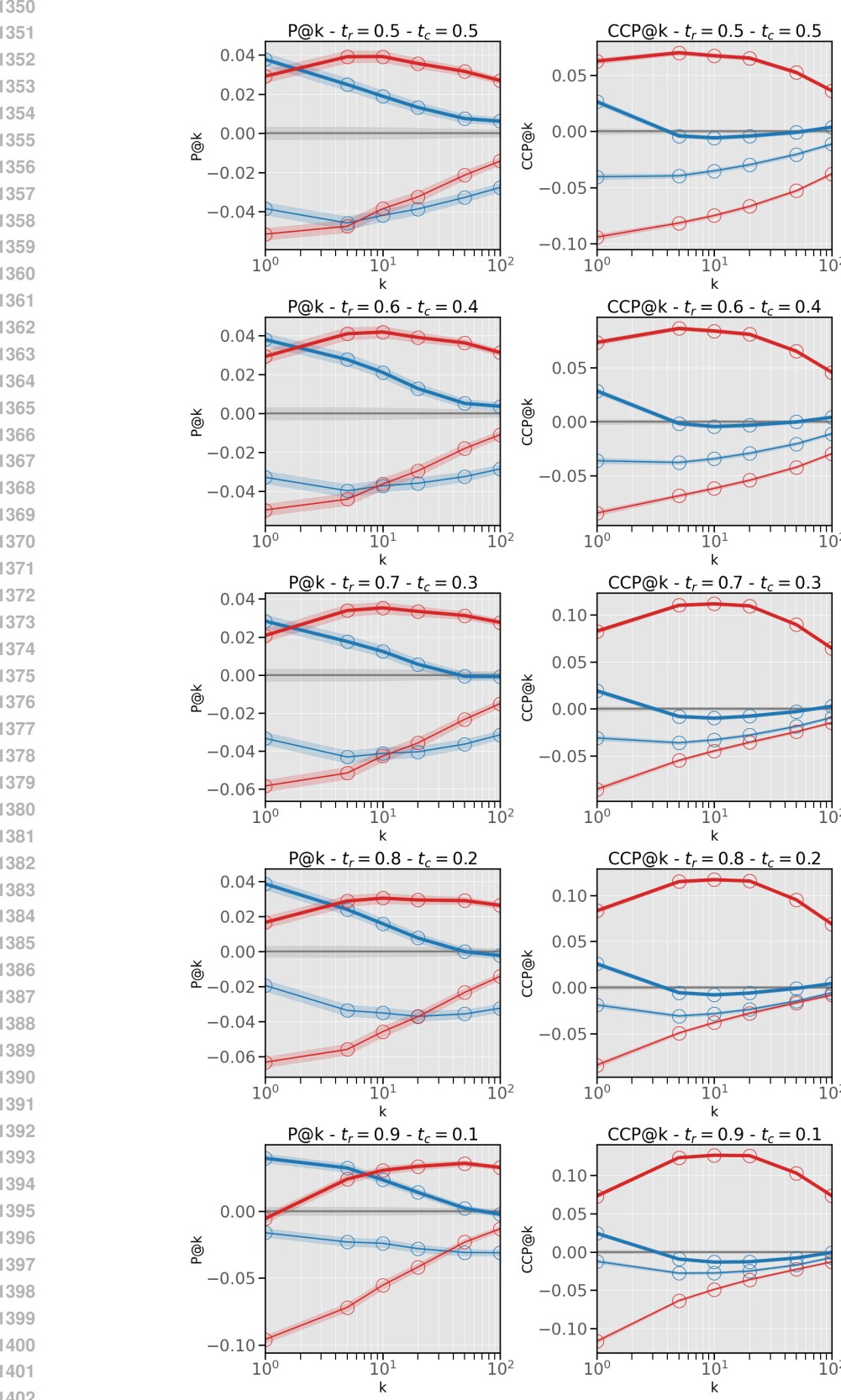

Figure 11: `bert-large-uncased`

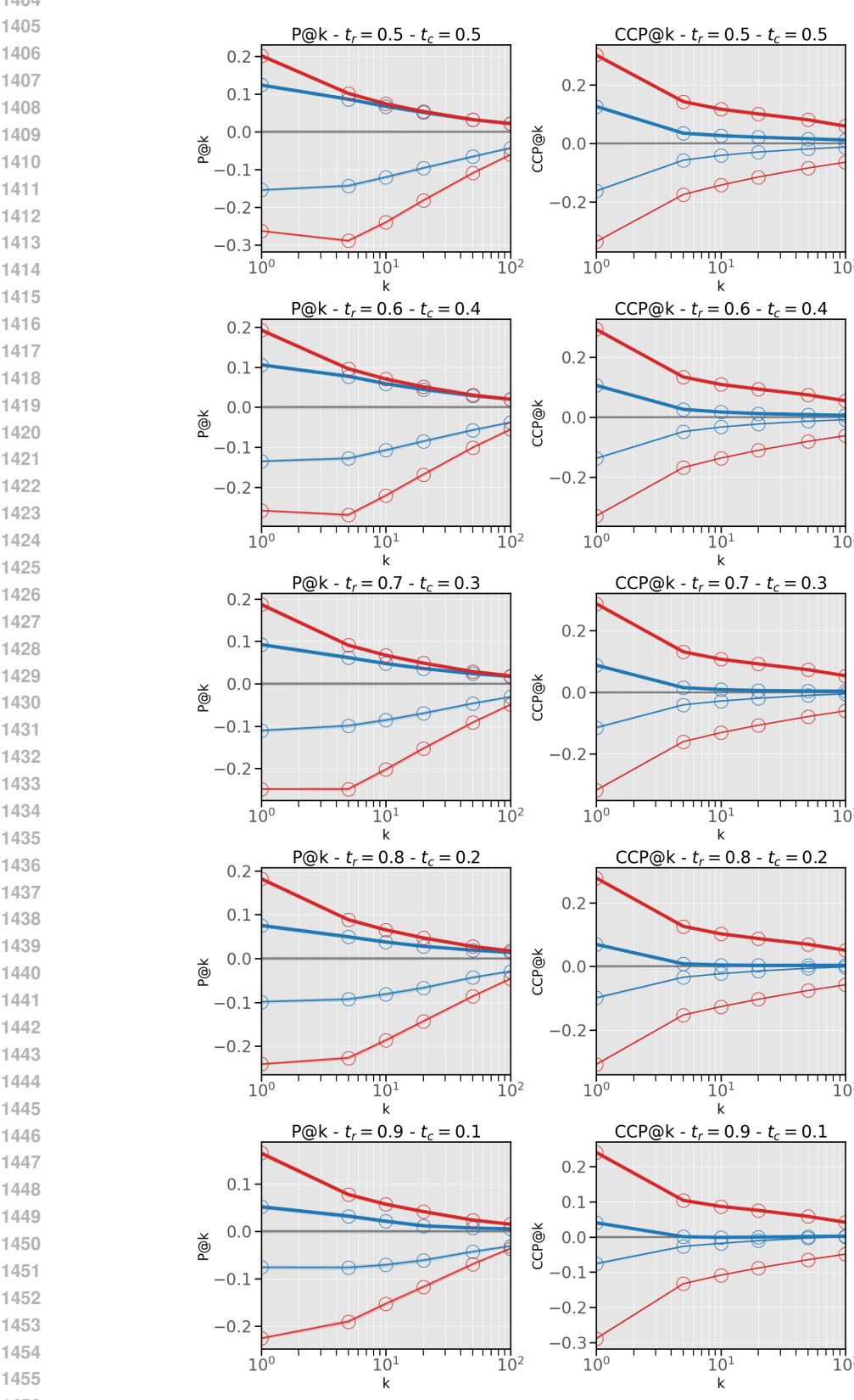

Figure 12: `opt-6.7b`

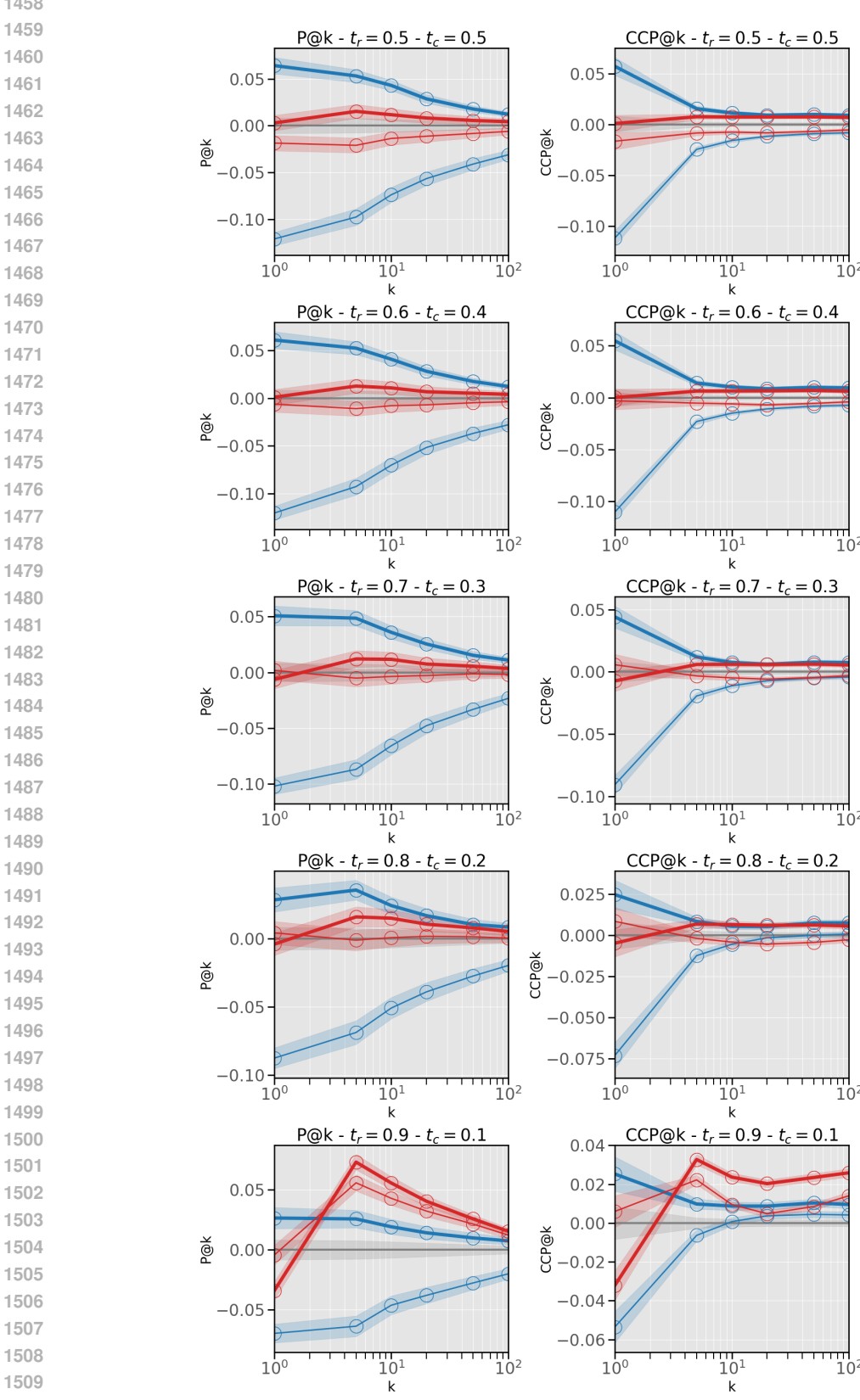

Figure 13: `gemma-2-9b`

