# OpenReview forum: "Qualifying Knowledge and Knowledge Sharing in Multilingual Models"
_ICLR.cc/2025/Conference — Submitted to ICLR 2025_

### Official Review · Reviewer_BLnS · 2024-10-28

**Soundness:** 3
**Presentation:** 2
**Contribution:** 2
**Rating:** 5
**Confidence:** 4

**Summary:**

In this paper, the authors provide a new perspective to refine the concept of "knowledge" by introducing the Concept Neurons and Relation Neurons. The authors also construct a multilingual version of the ParaRel dataset called mParaRel to analyze the language-agnostic knowledge neurons.

**Strengths:**

The analysis of knowledge storage and retrieval mechanisms for large language models is very important. This paper introduces concept neurons and relation neurons to study the knowledge mechanism of large language models, which helps to further explain large models. Besides, the authors create a dataset called mParaRel covering 10 languages, which helps to further analyze the internal mechanisms of multilingual large language models.

**Weaknesses:**

1. The authors should conduct more comprehensive experiments to demonstrate the accuracy of the localized concept neurons and relationship neurons. The stability or reliability of the experimental results.

2. Compared to their previous work, the authors extend the discovery of language-agnostic knowledge neurons from 2 languages to 10 languages, while this contribution is somewhat limited.

**Questions:**

1. Why not use the change rate of output probability as an evaluation metric, when manipulating concept neurons or relationship neurons? The change rate of output probability has been adopted by previous work (Dai et al. 2022).

2. Does the findings of this article apply to larger scale models? It would be better if a larger model could be analyzed, but it may be limited by computational resources.

---

> ### Author Response · Authors · 2024-11-16
>
> Thank you very much for your thoughtful review. We hope the following clarifications can help resolve your concerns.
>
> - *“The authors should conduct more comprehensive experiments to demonstrate the accuracy of the localized concept neurons and relationship neurons. The stability or reliability of the experimental results.”*
>
>     We approach knowledge neurons with skepticism, aiming to provide a broader view of the desired behavior of neurons with such properties. Few neurons exhibit a precise behavior with a clear connection to a concept or a relation, as shown in Figure 2. However, there is meaningful signal, which we test experimentally in Section 5.3 to evaluate if this natural classification aligns with model behavior. Our findings reveal the polyfunctional nature of neurons: we find stability across thresholds, across models (although it is clearer in some models such as bert-large and the most recent gemma-2-9b), and across languages for multilingual settings. Admittedly, we started with much skepticism about single neuron approaches, and found more stability than we expected, which we hope to report fairly. We would be happy however to report any area of doubts in the results in a revised version.
>
>
> - *“Compared to their previous work, the authors extend the discovery of language-agnostic knowledge neurons from 2 languages to 10 languages, while this contribution is somewhat limited.”*
>
>     Disclaimer: the work on 2 languages is not ours. We believe that analyzing more than two languages is not just an incremental step forward, but essential. Sharing across two languages collapses a range of possibilities: are neurons shared pairwise or are they shared with no or all languages at once? Is there more sharing with the dominant language (e.g., if English is more represented in the dataset)? Also, the probability that a neuron would end up being shared by two languages by pure chance is (much) higher than it be shared across 10 languages, hence this contributes to addressing the robustness issue raised above.By including 10 languages, we reveal that the different languages play symmetrical roles in the sharing of neurons, and that sharing happens way beyond pairs of languages.  Going way beyond what could have been a bias for, say, English then, this provides robust evidence of truly multilingual knowledge representation.
>
> To answer your questions:
>
> - *“Why not use the change rate of output probability as an evaluation metric, when manipulating concept neurons or relationship neurons? The change rate of output probability has been adopted by previous work (Dai et al. 2022).”*
>
>      We computed our results with relative probability changes as in Dai et al 2022, and found similar results. This measure is not perfect however. Specifically, there could be an important probability increase for the correct answer, say, without the obtained probability making the correct answer anywhere close to being eligible as the final prediction of the model. We therefore decided to report P@k and CCP@k in the main text, as it better captures the effect on behavior (rather than on mere activations). We would be happy to report the (similar) results using relative probability changes in the main text or in an appendix, or to provide the motivation for our choice more clearly in the revised version (or both).
>
> - *“Do the findings of this article apply to larger-scale models? It would be better if a larger model could be analyzed, but it may be limited by computational resources.”*
>
>     We extended our analysis to more recent and larger models, starting with BERT and expanding to LLaMA-2-7b and Gemma-2-9b. While computational resources limit testing even larger models, we analyzed the largest available to us to ensure the scalability of our findings. At this stage, we prioritized model diversity rather than model size (in part because this comes with more training diversity as well, see also discussion above). Since this may have been missed, as a similar remark was made by another reviewer, we can make these results more visible in the main text.

---

### Official Review · Reviewer_zSM3 · 2024-11-03

**Soundness:** 2
**Presentation:** 2
**Contribution:** 2
**Rating:** 3
**Confidence:** 4

**Summary:**

This paper investigated the knowledge representation mechanism of language models based on knowledge neuron. Specifically, by investigating knowledge neurons across different instantiations, prompts and languages, they find that 1) knowledge neurons exhibit different flavors, some encoding entity-level concepts, others with a more polysemantic role. 2) part of the knowledge neurons are shared by certain languages. 3) some knowledge neurons of certain LMs are shared between natural and unnatural prompts (specifically autoprompt).

**Strengths:**

- The findings and conclusions of this paper contribute to researchers' understanding of the knowledge storage mechanisms in LMs, thereby facilitating further in-depth research in the future.
- The proposed multilingual dataset can benefit future relevant studies.
- This paper offers several interesting perspectives for analyzing the knowledge mechanisms of LMs.

**Weaknesses:**

- My primary concern regarding this paper is the **reliability and generalizability of its conclusions**.
  - All analyses are based on the knowledge neuron (Dai et al. 2022), a gradient-based algorithm initially proposed for encoder-only models. However, the authors did not verify whether their conclusions hold with other knowledge attribution methods (e.g., ROME, SAE). This significantly limits the applicability of their corresponding conclusions.
  - In their analysis of Relation Neuron and Concept Neuron, the authors subjectively define the selection of relevant knowledge neurons through threshold values. Consequently, the identified knowledge neurons are significantly affected by the chosen thresholds, making it uncertain whether these neurons genuinely encode the information as claimed by the authors. To address this, **the authors should incorporate more intuitive and precise analytical methods to ascertain the reliability of the identified neurons**. For instance, they could investigate whether perturbing these neurons affects the model's expression of a specific concept without impacting other concepts.
  - For a probing study, the authors should incorporate more recent and larger models, considering that LLaMA-2 has already been released for over a year. Additionally, it would be beneficial to conduct a more in-depth analysis of the consistency and differences in conclusions across different models.
  - Regarding the authors' analysis of different prompt types, they considered only a single prompt search algorithm (AutoPrompt). However, since both AutoPrompt and KN are gradient-based methods, the reliability of the corresponding conclusions is questionable.

**Questions:**

See Weakness

---

> ### Author Response · Authors · 2024-11-16
>
> Thank you very much for your valuable review. Below, we address your key points:
> - *"All analyses are based on the knowledge neuron [...] the authors did not verify whether their conclusions hold with other knowledge attribution methods (e.g., ROME, SAE). This significantly limits the applicability of their corresponding conclusions."*
>
>     We limited our study to Knowledge Neurons due to their popularity and to illustrate our methodology. Extending the work to other attribution methods, such as ROME or SAE, would indeed be valuable. We actually believe that the field as a whole should take up on the task to compare these knowledge attribution methods. This is of interest to us, but beyond our core research question for this paper. If it had been done, and if it was found that several methods actually identify similar neurons, then there would be no need to do the comparison here. If they diverge, and one could be claimed to be superior, then our method will be readily available and could be applied to this one. Hence, independent progress on this orthogonal front would be immediately applicable to our work.  Here, we focus on methods that attribute knowledge at the neuron level, specifically because we thought that this scale of analysis requires further scrutiny: is that a relevant scale of analysis, is anything encode at the level of a neuron? To our surprise, our findings align with previous research and suggest that some aspects are indeed encoded at the level of a neuron, and that a neuron could in fact encode “polysemous” (a Knowledge Neuron may encode not only one thing, but multiple things). These findings prompted us to explore concept disentanglement using SAEs, which we hope to present in future work, and would provide more methods for the field. That said, we could include ROME and MEMIT to explore the level of consistency of these methods if the reviewer believes that it is a crucial aspect, although as we said it would be better done in an independent paper that compares these methods in the first place.
>
> - *"In their analysis of Relation Neuron and Concept Neuron, the authors subjectively define the selection of relevant knowledge neurons through threshold values. [...] For instance, they could investigate whether perturbing these neurons affects the model's expression of a specific concept without impacting other concepts."*
>
>     The thresholds are applied post-calculation of Knowledge Neurons, which is consistent with Dai et al. 2022. As we do not know a priori which neurons play specific roles, we performed an exhaustive study across varying thresholds. Figures 2 and 3 illustrate this comprehensive exploration, providing insights into the behavior and distribution of different neuron types. We agree with the second point nonetheless, and indeed we conducted boosting experiments (Section 5.3, Figure 3) to examine the reliability and impact of the identified neurons, testing hypotheses (i), (ii), and (iii) in Section 5.3. These experiments highlight both the challenges and the varying degrees of signal across models. For example models such as bert-large and gemma-2-9b have stable positive results. We also note that we conducted these boosting experiments in the standard way, although using metrics which are more specific (see also response to reviewer 4). Specifically, we do not report simply whether the probability of the correct response rises, but we report whether the correct answer becomes the predicted answer. A positive result with this more conservative measure shows that the boosting effects are specific (it did not simply raise any vaguely relevant answer, but raised the target answer more than others).
>
> - *"For a probing study, the authors should incorporate more recent and larger models, [...] analysis of the consistency and differences in conclusions across different models."*
>
>     We agree entirely that using modern and large models is important! Hence, starting with BERT for continuity with prior work, we expanded to larger and newer models, the results do include LLaMA-2-7b and Gemma-2-9b (released ~3 months ago), see Appendix. While we still aspire to analyze even larger models too, computational constraints have pushed us towards model diversity rather than size at this stage. To avoid the possibility of missing this, we can include a summary table comparing model behaviors and findings in the main text.
>
> - *"Regarding the authors' analysis of different prompt types,[...] both AutoPrompt and KN are gradient-based methods, the reliability of the corresponding conclusions is questionable."*
>
>     This is a subtle concern! While they use the same gradient-based method, AutoPrompt optimizes prompts, while gradients are used differently to identify KN. We cannot pinpoint a specific mechanism or circularity that this could introduce which would artificially create the results as we see them (sharing between AutoPrompts, but also across languages).

---

> > ### Comment · Reviewer_zSM3 · 2024-11-22
> >
> > Thanks for your detailed reply.
> >
> > Overall, this paper primarily comprises two key factors: the analytical approach towards knowledge sharing and the corresponding conclusions derived from the analysis. To be frank,I think that the analytical method in this paper does not have much technical innovation. **Therefore, in my opinion, the core contribution of this paper should lie in its conclusions. In such a context, the reliability and generalizability of the conclusions are crucial to the paper's value**.
> > If the corresponding conclusions are limited to "the gradient-based knowledge neuron", perhaps the title would be more appropriately phrased as "The Sharing of Knowledge Neurons."
> >
> >
> > > The thresholds are applied post-calculation of Knowledge Neurons, which is consistent with Dai et al. 2022. As we do not know a priori which neurons play specific roles, we performed an exhaustive study across varying thresholds.
> >
> > I am referring to the threshold used to distinguish between concept neurons and relation neurons, rather than the threshold used to identify knowledge neurons.
> >
> > > we conducted boosting experiments (Section 5.3, Figure 3) to examine the reliability and impact of the identified neurons,
> >
> > 1. Can you further explain the distinct behaviors between these models? It appears that the model's behavior does not fully align with expectations.
> >
> > 2. Moreover, I did not fully understand the hypothesis underlying this experiment. Why does this experiment demonstrate that the identified relation neurons and concept neurons are linked exclusively to specific relations and concepts, without affecting other relations and concepts?
> >
> > > We cannot pinpoint a specific mechanism or circularity that this could introduce which would artificially create the results as we see them (sharing between AutoPrompts, but also across languages).
> >
> > I believe it is necessary to include additional algorithms for prompt-tuning.

---

> > > ### Author Response · Authors · 2024-11-25
> > >
> > > Thank you. We would be happy to change the title in any way appropriate. To recap: we use an imperfect knowledge detection method, and yet detect specific neurons of various degrees of generality (concept vs relation neurons, see below), and unprecedented multilingual sharing (based on a new dataset for the community) in some models but not all (see below).
> > >
> > > *“I am referring to the threshold used to distinguish between concept neurons and relation neurons, rather than the threshold used to identify knowledge neurons.”*
> > >
> > > We agree. Let us clarify: we explored large threshold ranges both for the determination of knowledge neurons and for the distinction between the concept vs relation neurons. In fact, it is part of the method to look at all possible thresholds. No major variation based on the choice of threshold was found.
> > >
> > > *“Can you further explain the distinct behaviors between these models? It appears that the model's behavior does not fully align with expectations.”*
> > >
> > > Exactly. Some models don't behave as expected. This either indicates that different architectures represent information differently at the neuronal level (strong interpretation), or a weakness in the knowledge neuron method (safer interpretation). This is part of our conclusion: KNs are a crude way of approaching the problem of knowledge in LLMs. However, they do deliver signal, to our surprise, and thus offer a reasonable proxy to measure, e.g., information sharing in the multilingual study.
> > >
> > > *“Moreover, I did not fully understand the hypothesis underlying this experiment. Why does this experiment demonstrate that the identified relation neurons and concept neurons are linked exclusively to specific relations and concepts, without affecting other relations and concepts?”*
> > >
> > > Thank you for asking, this matters a lot, and we can see why it required taking action. Two answers then.
> > > 1. The usual metric for boosting is impact on the probability of correct answer, which indeed would not show specificity. For this reason, we report Precision at rank k and Correct Category Prediction metrics. Effects there ensure that the boost to the correct answer overcomes any boost for other answers.
> > > 2. We have carried out a new control experiment in which we boost the KNs of a target instantiation, while testing either the target question (as before) or question from another instantiation (e.g. we test the effect of boosting KNs of “London” when we prompt “The capital of France is”). Results are very clear: performance is massively degraded in the latter case specifically (P@k: +2.5% in the aligned case vs -1% in the misaligned case). We run these tests both for concept neurons (as explained) and with relation neurons (using the question from another *relation* then, not only instantiation). Thanks to your suggestions, we will add these new results to the main part of the paper.
> > >
> > > *“I believe it is necessary to include additional algorithms for prompt-tuning.”*
> > >
> > > The results on Autoprompt are an independent section in the paper, a control that we believe is very rare, and which leads to a new dataset delivered to the community. Depending on resources, we will try to reproduce these with other prompt-tuning methods. Methods which are not based on gradient may come across as creating fewer risks of a confound, (a risk that we need to pinpoint more). But on the other hand they often yield prompts closer to their natural language counterparts (e.g., shared words), which could be less fruitful and the source of more immediate problems for interpreting parallelism in the behaviors.
> > >
> > > Please let us know if you still have any additional concerns or questions.

---

### Official Review · Reviewer_KXuV · 2024-11-04

**Soundness:** 3
**Presentation:** 3
**Contribution:** 3
**Rating:** 5
**Confidence:** 3

**Summary:**

This paper explores how multilingual language models encode and share knowledge across languages. Specifically, the authors categorize Knowledge Neurons (KNs) into two subtypes: relation neurons and conceptual neurons. They investigate to what extent these neurons exist, their contributions to knowledge retrieval, and their impact on downstream tasks. The paper introduces mParaRel, a multilingual variant of ParaRel, to facilitate experiments across ten languages. The authors find that while most KNs do not adhere to a clearly defined role, some KNs do serve specific functions. Their findings suggest that many KNs are shared across languages, indicating a partially language-agnostic retrieval mechanism in multilingual models.

**Strengths:**

1. Differentiating conceptual neurons and relation neurons is a meaningful theoretical advancement in studying knowledge representation.

2. The mParaRel dataset is a valuable contribution to multilingual probing research.

3. The experiments provide sufficient support for the paper's central claim regarding the individuality and shareability of knowledge neurons across languages.

**Weaknesses:**

1. The contribution of the paper is not novel or different from the lines of work in multilingual knowledge probing, where they all indicate that there exist language-agnostic and language-specific neurons such as [1][2][3]. None of the works were cited or discussed here.

2. The authors rely on threshold-based methods to classify neurons as either conceptual or relational, but variations in thresholds produce different neuron classifications. This makes the claim unrobust: line 208 "When the thresholds become more restrictive, the number of neurons with well-defined roles decreases". There needs to be more effort in verifying that the relation neurons are indeed representing perform identical relational functions across languages.

[1] Kojima, Takeshi, et al. "On the Multilingual Ability of Decoder-based Pre-trained Language Models: Finding and Controlling Language-Specific Neurons." NAACL (2024).

[2] Tang, Tianyi, et al. "Language-specific neurons: The key to multilingual capabilities in large language models." ACL (2024).

[3] Wang, Weixuan, et al. "Sharing Matters: Analysing Neurons Across Languages and Tasks in LLMs." arXiv preprint arXiv:2406.09265 (2024).

**Questions:**

What are your insights regarding the differences in knowledge-sharing levels among various language pairs? Figure 4 indicates that Spanish shares more knowledge with Danish and Dutch than with French or Italian, which belong to the same Latin family. Do you observe any cross-model similarities related to this finding?

---

> ### Author Response · Authors · 2024-11-16
>
> Thank you very much for your thoughtful review. We’d like to clarify some points you raised:
> - *"The contribution of the paper is not novel or different from the lines of work in multilingual knowledge probing, where they all indicate that there exist language-agnostic and language-specific neurons such as [1][2][3]. None of the works were cited or discussed here."*
>
>     Thank you for highlighting these references. Our method indeed demonstrates similar findings but in a specific context: neurons that are selected as relevant for knowledge retrieval. We aim not only at asking whether neurons are shared across languages, but also what neurons are shared, depending on their functions in the overall task. Are the shared neurons involved in getting the task right (hence activating for all instances of the task), or specific for some facts, or for some entities?. Our results go beyond the references by detailing both the sharing and functionality, and how this happens across multiple languages. You are absolutely right that we should discuss these references, and we will.
>
>
> - *"The authors rely on threshold-based methods to classify neurons as either conceptual or relational, but variations in thresholds produce different neuron classifications. This makes the claim unrobust: line 208 'When the thresholds become more restrictive, the number of neurons with well-defined roles decreases'. There needs to be more effort in verifying that the relation neurons are indeed performing identical relational functions across languages."*
>
>     We fully agree that using thresholds introduces fragility. To address this, we calculated our results across a range of thresholds. Figures 2 and 3 show the results for various choices of thresholds: yes, more restrictive thresholds exclude more neurons (by definition), but the main results (about the type of shared neurons and amount of sharing across languages) are stable independently of a specific choice of a threshold.
>
> - *“What are your insights regarding the differences in knowledge-sharing levels among various language pairs? Figure 4 indicates that Spanish shares more knowledge with Danish and Dutch than with French or Italian, which belong to the same Latin family. Do you observe any cross-model similarities related to this finding?”*
>
>     We explored potential correlations between the number of shared neurons and various linguistic characteristics. Language family did not produce any clear pattern (coherent with the fact about Spanish mentioned above); syntactic distance (à la Philippy et al., 2023) was a marginal predictor of the amount of neuron sharing (p=.05). We could report these results, even though they are only marginally significant at this stage. Differences could also emerge due to differences in training sets (some may be more parallel than others across pairs of languages, information to which we did not have access to). We agree that it is an important question, and we hope our work reveals its importance (it was not a possible question to ask in the work mentioned by another reviewer where only two language pairs were considered) and that our work makes it possible to address it (both the current results and the introduction of the new dataset mParaRel). We welcome any further suggestion for other measures of distance across languages that could be a good predictor of the degree of neuron sharing.
>
>     We tested two multilingual models (only two because of computational frugality at this stage, since the demands increase as we test each model with 10 languages). The pattern you picked up on (Spanish vs Romance/non-Romance languages) was observed for LLaMA-2-7b, but not for bert-base-multilingual. We do not know if it is due to their different architectures, or difference in their training sets at the moment. Again, we hope that our contribution makes it possible to tackle these important questions.

---

### Official Review · Reviewer_6Nqi · 2024-11-06

**Soundness:** 3
**Presentation:** 3
**Contribution:** 4
**Rating:** 8
**Confidence:** 4

**Summary:**

This paper presents a valuable investigation into knowledge representation within pre-trained language models (PLMs), specifically exploring how these models encode and retrieve factual knowledge. The authors introduce a fine-grained typology of knowledge that separates various aspects, such as entity concepts, relationships, and prompt structures. By examining the Knowledge Neuron (KN) framework across multiple PLMs, they reveal that not all neurons are monosemantic; rather, a significant portion exhibit polysemantic behavior, with only certain neurons specialized for specific concepts or relationships.

The study’s empirical findings, derived from a range of models (e.g., BERT, Llama 2, and mBERT) and 10 languages, uncover substantial overlap in knowledge representation, suggesting a cross-linguistic, language-agnostic retrieval system. The authors also introduce the mParaRel dataset, an extension of ParaRel in 10 languages, enhancing the scope of knowledge retrieval evaluation. These contributions imply that language models may benefit from shared multilingual training, offering insights into efficient expansion to new languages.

Overall, the paper provides a thoughtful critique of neuron-level attribution techniques, indicating that future PLM training could optimize for unique linguistic characteristics rather than relearning factual knowledge for each language.

**Strengths:**

Originality: This paper is highly novel, introducing a new typology for analyzing knowledge in PLMs that challenges the Knowledge Neuron hypothesis of monosemanticity. By revealing polysemantic behaviors in neurons, the study provides a nuanced view of knowledge representation, paving the way for a broader understanding of how PLMs encode concepts and relationships.

Quality: The study is well-executed, with a solid experimental design across a variety of PLMs and languages. The release of the multilingual mParaRel dataset further enhances the quality and replicability of this work.

Clarity: The paper is clearly organized, especially in detailing the methods and experimental results, making the findings accessible to readers.

Significance:
This work has considerable impact, as it suggests that knowledge in multilingual models may be stored in a language-agnostic manner, meaning retraining for each language might be unnecessary. This insight holds potential for more efficient multilingual model training, especially for low-resource languages, and offers a promising direction for leveraging neuron-level analysis in PLMs to improve model interpretability and efficiency.

**Weaknesses:**

P9 L468: The formula $(\text{number of languages})^{-\alpha}$ is introduced, but the value and explanation of $\alpha$ are not provided. It would help to describe this parameter to make the formula more rigorous.

P9 L468-469: Assuming $\alpha$ is between 0 to 1, both $(\text{number of languages})^{-\alpha}$ and $p^{\text{number of languages}}$ decay as the number of languages increases. How can we be confident that the curve shown in Fig. 4c follows the former form and not the latter? A brief explanation of how the curve was fitted to these two functions would be helpful, as it directly impacts one of the major conclusions that KNs are multilingual.

**Questions:**

P4 L163-164: "we apply our analysis to the earliest of these methods by way of illustration." Could you please specify which is "the earliest" of these methods?

---

> ### Author Response · Authors · 2024-11-16
>
> Thank you very much for your thoughtful and detailed review! We very much appreciate your interest in the work, and are happy to see that we share the same vision about the benefits of looking at multilingual shared neurons. We hope that this work and the dataset could help the community move forward in that direction. We will make the sentence you mention clearer (we will mention Knowledge Neurons explicitly), and we will add the information about the fitting of the curve.

---

> > ### Comment · Reviewer_6Nqi · 2024-11-27
> >
> > Thanks to the authors for the reply. Looking forward to a more perfected version of the paper.

---

### Public Comment · ~Constanza_Fierro1 · 2024-11-19
**Nitpick: name of the dataset**

This is a cool work! Just a small suggestion, maybe it would be a good idea to change the name of the dataset, as we also developed a translation of ParaRel a few years ago and named it mParaRel (https://aclanthology.org/2022.findings-acl.240/) so it could get confusing...

---

> ### Author Response · Authors · 2024-11-19
>
> Thank you very much for your comment! We didn't know the name was taken, sorry! We will think of a new name.

---

### Author Response · Authors · 2024-11-28
**Submission of Rebuttal Revision**

Thank you for the valuable feedback provided during the review process. We have uploaded a revised version of our paper addressing the key concerns raised. Below is a summary of the major updates:

- Clarified Lines 163-164 for improved readability and precision.
- Fitted the curve in Section 6.2 and added the fitted curve along with the baseline to the corresponding graph.
- Included the proposed references and discussed them in the Related Work section.
- Expanded the discussion on the use of thresholds in Section 4, incorporating feedback from reviewers.
- Provided clearer explanations for the motivations behind our metric choices in Section 5.3.
- Added a summary table comparing model behaviors against expectations for the Boosting experiment (Appendix).
- Discussed the potential bias of using gradient-based AutoPrompt and Knowledge Attribution methods in Section 6.3.
- Included a discussion emphasizing the importance of studying more than two languages in the Related Work.
- Renamed our multilingual dataset as required.
- Added the discussed control experiment for the Boosting experiment (Appendix).

We hope these revisions address the concerns raised and improve the clarity and quality of the paper. As always, we welcome any further questions or comments.

---

### Meta-Review · Area_Chair_iEAV · 2024-12-19

**Metareview:**

**Summary**

This paper aims to build on the notion o knowledge neurons in order to find specific neurons where interlingual knowledge is stored. Two kinds of knowledge neurons are introduced: conceptual neurons adn relation neurons. Thee existence of these neurons is experimentally searched and some specific neurons are spotted, albeith the majority of neurons do not have a specific role.

**Strengths**

- Introducing a classification of knowledge neurons
- Proving that this classification fits existing neurons

**Weaknesses**

- This paper is an observation paper that, besides the possibility to retreive these neurons, it does not give any possible way to modify these neurons in order to change the global behavior of the network.
- A comparison with alternative methods like ROME and MEMIT is missing

**Final remarks**

Knowledge neurons are interesting abstractions of a very complex phenomena. The idea that only a bunch of neurons is responsible for a specific behavior is very interesting. However, attribution is not sufficient to demostrate that these knowledge neurons exist.

**Additional Comments On Reviewer Discussion:**

During the discussion phase, the authors failed to convince the most unconvinced reviewer. The reviewer was partially open to the discussion, but they were not willing to change their mind.

---

### Decision · Program_Chairs · 2025-01-22

Reject